# The Preventive Effect of Dietary Antioxidants against Cervical Cancer versus the Promotive Effect of Tobacco Smoking

**DOI:** 10.3390/healthcare7040162

**Published:** 2019-12-13

**Authors:** Masafumi Koshiyama, Miwa Nakagawa, Ayumi Ono

**Affiliations:** 1Department of Women’s Health, Graduate School of Human Nursing, The University of Shiga Prefecture, Shiga 522-8533, Japan; 2School of Human Nursing, The University of Shiga Prefecture, Shiga 522-8533, Japan; nakagawa.miw@nurse.usp.ac.jp (M.N.); ono.a@nurse.usp.ac.jp (A.O.)

**Keywords:** cervical cancer, dietary oxidant, tobacco smoking

## Abstract

Uterine cervical cancer is the fourth most common cancer in women, and its etiology has been recognized. High-risk human papilloma virus (HR-HPV) infection induces an opportunity for malignant transformation. This paper discusses the current issues based on a review of the literature and compares the impact of the dietary and nutrient intake to the impact of tobacco smoking on cervical cancer development. The important roles of diet/nutrition in cervical cancer are as prophylaxis against HR-HPV infection. Antioxidant vitamins can inhibit the proliferation of cancer cells, stabilize the p53 protein, prevent DNA damage, and reduce immunosuppression. In contrast, tobacco smoking not only causes DNA adducts and strand breaks, but it independently causes an increased viral load in HR-HPV-infected cells. Tobacco smoking induces the heightened expression of E6 and E7 and can inhibit the immune system response to HPV. What happens when two materials, which have opposite effects on cervical cells, are taken in at the same time? The negative effects of tobacco smoking may be stronger than the positive effects of vitamins, vegetables, and fruits on the regression of cervical disease such as cervical intraepithelial neoplasia (CIN). A relatively low intake of vitamins, vegetables, and fruits in combination with tobacco smoking was most associated with a high incidence of cervical neoplasia.

## 1. Introduction

Cervical cancer is the fourth most frequent cancer in women with an estimated 570,000 new cases in 2018 [1]. The causal relationship between high-risk human papillomavirus (HR-HPV) infection and cervical cancer development has been well documented (up to 99.7%) [2].

Exposure to tobacco smoke is associated with the development of multiple cancers [3,4]. In 2004, the International Agency for Research on Cancer also classified tobacco smoking as a cause of cervical cancer [5]. It is reported that 11.8% of cervical cancer deaths are attributable to smoking [6]. Thus, tobacco smoking has been associated with the progression of cervical dysplasia and cancer. In women who smoke tobacco, the risk of developing cervical cancer is up to two times higher than that in women who do not smoke tobacco [7].

In contrast, the important role of antioxidant nutrients/diets, such as vitamins, fruits, and vegetables, in preventing the development of cervical cancer has received a lot of attention during recent decades [8]. The important roles of dietary antioxidants in cervical cancer are as prophylaxis against HPV infection.

This paper reviews the current issues in the relevant literature and the effects of the antioxidant diets/nutrients and smoking on cervical cancer, and it discusses the points to be noted.

## 2. HPV Infection and Cervical Carcinogenesis

HPV types 6, 11, and 42 are associated with benign genital warts and low-grade cervical intraepithelial neoplasia (CIN), while HR-HPV such as HPV types 16, 18, and (less frequently) 31, 33, and 35 have been identified as an infectious agent to cause high-grade precursor lesions and the majority of cervical cancers [9,10]. Persistent HR-HPV infection has been recognized as a necessary step in the progression of CIN, which is graded from 1 to 3 depending on the degree of epithelial abnormality, and cervical cancer. Among them, HPV 16 and 18 are identified in ≥ 70% of cervical cancer cases [11,12]. Especially, HPV 16 has the ability to evade the immune system even in immune-competent individuals [13]. The main factors in the malignant transformation are E6 and E7 proteins [14]. E6 has the ability to induce the formation of a complex with ubiquitin ligase and p53, resulting in inhibition of p53-mediated apoptosis [15,16]. E7 has the ability to bind to the retinoblastoma (Rb) protein, resulting in inhibition of Rb functions such as cell cycle regulation [17].

Exposure at a young age also increases the risk of persistent HR-HPV [18]. Thus, HR-HPV infection is most common in young women 18–30 years old, with prevalence sharply declining after age 30. In addition, the metaplastic activity of the cervix increases during puberty and the first pregnancy. Therefore, the median age of HPV-associated cervical cancer diagnosis is 49.

## 3. Effects of Oxidative Stress and HPV-Related Carcinogenesis

A close link between chronic inflammation, oxidative stress, and carcinogenesis has been reported [14,19]. Inflammation has been regarded as a potential promoter of carcinogenesis in reactive oxygen species (ROS) production [20]. Inflammation due to HPV infection leads to the release of interleukin (IL)-1, IL-6, tumor necrosis factor-α, and interferon gamma, resulting in the formation of ROS [21]. Inflammation in HR-HPV-infected cells also causes a decrease in the level of antioxidants [22].

HR-HPV integration is facilitated by the generation of DNA damage/double strand breaks (DSBs) [23], which causes DNA oxidation. Oxidative damage of the genome is associated with epigenetic alterations [24]. The major product of DNA oxidation is also correlated with increased HPV infection, viral–host integration, and dysplasia development [25]. Thus, apoptosis is hindered by the disruption of many regulatory pathways, which results in an altered cellular proliferation [26].

## 4. The Effects of Dietary Antioxidants on the Development of Cervical Cancer

Antioxidants can act as efficient scavengers of free radicals and oxidants to prevent DNA damage [27]. If antioxidants cannot neutralize free radicals and oxidants, inflammation caused by HR-HPV infection on the cervix leads to extensive damage to DNA [28].

Antioxidant vitamins (e.g. vitamins A, C, D, and E) can inhibit the proliferation of cancers [29,30], stabilize the p53 protein [31], prevent DNA damage, and reduce immunosuppression [32,33]. 

Most recently, we reviewed the recent clinical studies of dietary oxidants on development of cervical cancer [34]. The following were established in the results. Dietary antioxidants may have different abilities in the natural history of cervical disease development associated with HR-HPV infection. Vitamin A (retinol), vitamin D, and papaya might be more effective in preventing early dysplasia, CIN 1. In contrast, vitamin E and lycopene might be more effective in preventing late dysplasia, CIN 2/3. Both fruits and carotenoids might widely prevent HPV infection, CIN 1–3, and cervical cancer. Vegetables might prevent HPV infection and CIN 1–3, except for cervical cancer.

## 5. Smoking and Cervical Carcinogenesis 

Wei et al. suggested that in HR-HPV-infected cells, tobacco smoking not only causes DNA adducts and strand breaks [3], but it also causes an increase in the viral load [6]. Tobacco smoking induces heightened E6 and E7 expression that has further adverse effects on cell cycle control, DNA damage repair, and protective apoptosis; thus, cervical cells continue to accumulate mutations that permit malignant transformation [6]. In addition, the immune suppression that occurs in the context of tobacco exposure can permit HR-HPV-infected cells to survive [35].

In a clinical study of 1007 women, the risk of HPV infection in smoking women was reported to be 1.905 times greater than that in nonsmoking women (odds ratio (OR) 1.905, *p* < 0.05) [36]. Furthermore, the risk of CIN 2–3 or cervical carcinoma in smoking women was reported to be 1.642 times greater than in nonsmoking women (OR 1.642, *p* < 0.05). In a population-based case-control study of 137 women with CIN 2–3 and 253 healthy aged-matched women, smoking was found to be associated with CIN 2–3 (OR 2.6, *p* < 0.001) [37]. This smoking effect was dose-dependent (*p* = 0.002). In a case-control study of 480 patients with cervical cancer and 797 population controls in the United States, a two-fold increase in risk was observed in patients who smoked 40 or more cigarettes per day and those who had smoked for ≥ 40 years [38]. The data from a meta-analysis of eight case-control studies also suggested that the incidence of cervical cancer among smokers was increased by 42%–46%, even after controlling for age and number of sexual partners [39]. 

Passive smoking is also associated with a risk of developing cervical cancer. In a case-control study with 177 cervical cancer cases and 177 controls, females with sexual partners who were smokers showed an increased risk of developing cervical cancer (OR 2.77, *p* < 0.001, and adjusted OR 3.15, *p* = 0.001) [40]. In a meta-analysis of 14 eligible studies, which included a total of 384,995 participants, the pooled OR of passive smoking for the risk of cervical cancer was 1.70 [41]. 

## 6. Discussion

During HR-HPV infection, E6 has the ability to inhibit the function of p53 [15,16], and E7 has the ability to bind to the retinoblastoma (Rb) protein. HR-HPV-infected cells involving these oncoproteins show proliferation and malignant transformation. Moreover, inflammation due to HPV infection causes a decrease in the level of antioxidants [22]. Thus, dietary antioxidant intake can stabilize the p53 protein [31] and prevent DNA damage and cancer development. 

In contrast, in HR-HPV-infected cells, tobacco smoke not only causes DNA adducts and strand breaks [3], but it also causes an increased viral load [6]. This induces an increase in the expression of E6 and E7. In addition, tobacco smoking can inhibit the immune system response to HPV, especially the activities of T helper lymphocytes, natural killer cells, and immunoglobulin E [42]. The expression of programmed death-1 (PD-1) is also observed in 47.0%–60.8% of cervical cancer patients [43,44,45]. Furthermore, active tobacco smoking or a history of tobacco smoking during radiation therapy for cervical cancer is associated with unfavorable disease-free and overall survival outcomes [46]. In brief, tobacco smoking can accelerate the development of cervical cancer, while antioxidant intake can suppress the development of cervical cancer.

Individuals can easily be exposed to tobacco smoke through passive smoking and consume dietary/nutrient antioxidants such as vitamins, vegetables, and fruits in the environment. What happens when two materials, which have opposite effects on cervical cells, are taken in at the same time? Fujii et al. reported some very interesting findings [47]. In nonsmoking patients with low-grade cervical abnormalities, disease regression was significantly associated with the serum levels of zeaxanthin/lutein (hazard ratio (HR) 1.25, *p* = 0.024). However, this benefit was abolished in smokers. Tomita et al. also reported that a low intake (≤ 39 g) of dark-green and deep-yellow vegetables and fruits without tobacco smoking was associated with a lower incidence of CIN 3 development (OR 1.14) in comparison to smokers with higher intakes of dark-green and deep-yellow vegetables and fruits (≥ 40 g; OR 1.83) [48]. In addition, the OR for the joint exposure of tobacco smoking and lower intake of vegetables and fruits was greater (OR 3.86, *p* < 0.001), compared with nonsmokers, with a higher rate of HR-HPV-infected cells. In patients with cervical disease, the negative effects of tobacco smoking may be stronger than the positive effects of vitamins, vegetables, and fruits. A relatively low intake of vitamins, vegetables, and fruits in combination with tobacco smoking was most associated with a high incidence of cervical neoplasia. In particular, young patients with cervical disease should, therefore, both stop any active smoking and increase their intake of dietary/nutrient antioxidants, while also being careful to avoid passive smoking.

## 7. Conclusions

In patients with cervical disease, the negative effects of tobacco smoking may have a greater impact than the beneficial effects of dietary antioxidant intake on cervical cancer development. Therefore, patients, in particular young patients with cervical disease, should stop any active smoking, and avoid passive smoking, while also increasing their intake of vitamins, vegetables, and fruits.

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
