# Peer review of "The Preventive Effect of Dietary Antioxidants against Cervical Cancer versus the Promotive Effect of Tobacco Smoking"

_healthcare, 2019, doi:10.3390/healthcare7040162_

Round 1
Reviewer 1 Report
needs more literature research on mechanisms of the many different potentially carcinogen types of hpv
also didnt mention the difference in the cervical lesion from hpv infection like CIN and low/high grade . also patients age plays important role, as low grade lesions in young females are more vulnerable and may be eliminated by the patients immune system
Author Response
Referee1
Comments and Suggestions for Authors
needs more literature research on mechanisms of the many different potentially carcinogen types of hpv also didnt mention the difference in the cervical lesion from hpv infection like CIN and low/high grade . also patients age plays important role, as low grade lesions in young females are more vulnerable and may be eliminated by the patients immune system
èThank you very much for your nice comments. Yes, we inserted the following sentences in 2. HPV infection and cervical carcinogenesis of text.
HPV types 6, 11 and 42 are associated with benign genital warts and low- grade cervical intraepithelial neoplasia (CIN), while HR-HPV such as HPV types 16, 18 and less frequently 31, 33 and 35 have been identified as an infectious agent to cause the high-grade precursor lesions and the majority of cervical cancers [9, 10].
Persistent HR-HPV infection has been recognized as a necessary step in the progression of CIN which is graded from 1 to 3 depending on the degree of epithelial abnormality, and cervical cancer. Among them, HPV 16 and 18 are identified in > 70% of cervical cancer cases [11, 12]. Especially, HPV 16 has the ability to evade the immune system even in the immune competent individuals [13].
Exposure at a young age also increases the risk of persistent HR-HPV [18].
Thus, HR-HPV infection is most common in young women 18-30 years old, with prevalence sharply declining after age 30. In addition, the metaplastic activity of the cervix increases during puberty and the first pregnancy. Therefore, the median age of HPV-associated cervical cancer diagnosis is 49.
Reviewer 2 Report
This is a significant mini review describing the relationship between the benefical effect of dietary antioxidants and the negative effect of tabacco smoking on the induction of uterine cervical cancer in women. The manuscript is well written in an engaging and lively style and, in my opinion, it is appropriate to our readership.
I have onlytwo minor comments that should be corrected or considered before publication:
Figure 1: The quality of this figure can be improved. Also, please delete "Figure 1" from the figure! The sentence, "What happens when two materials, which have opposite effects on cervical cells, are taken in at the same time?" has been repeated in the abstract, discussion and conclusion. I realize that the authors want to stress the concept but, please, avoid so many repetitions.Author Response
Referee2
Comments and Suggestions for Authors
This is a significant mini review describing the relationship between the benefical effect of dietary antioxidants and the negative effect of tabacco smoking on the induction of uterine cervical cancer in women. The manuscript is well written in an engaging and lively style and, in my opinion, it is appropriate to our readership.
I have onlytwo minor comments that should be corrected or considered before publication:
Figure 1: The quality of this figure can be improved. Also, please delete "Figure 1" from the figure! The sentence, "What happens when two materials, which have opposite effects on cervical cells, are taken in at the same time?" has been repeated in the abstract, discussion and conclusion. I realize that the authors want to stress the concept but, please, avoid so many repetitions.
Thank you very much for your good comments. Yes, we delete Figure 1.
In addition, we delete "What happens when two materials, which have opposite effects on cervical cells, are taken in at the same time?" in Conclusion.
